# Simplicial models of social contagion

Iacopo Iacopini [1,2], Giovanni Petri[3,4], Alain Barrat [3,5] & Vito Latora [1,2,6,7]

Complex networks have been successfully used to describe the spread of diseases in populations of interacting individuals. Conversely, pairwise interactions are often not enough to characterize social contagion processes such as opinion formation or the adoption of novelties, where complex mechanisms of influence and reinforcement are at work. Here we introduce a higher-order model of social contagion in which a social system is represented by a simplicial complex and contagion can occur through interactions in groups of different sizes. Numerical simulations of the model on both empirical and synthetic simplicial complexes highlight the emergence of novel phenomena such as a discontinuous transition induced by higher-order interactions. We show analytically that the transition is discontinuous and that a bistable region appears where healthy and endemic states co-exist. Our results help explain why critical masses are required to initiate social changes and contribute to the understanding of higher-order interactions in complex systems.

[1] School of Mathematical Sciences, Queen Mary University of London, London E1 4NS, UK. [2] The Alan Turing Institute, The British Library, London NW1 2DB, UK. [3] ISI Foundation, Via Chisola 5, 10126 Turin, Italy. [4] ISI Global Science Foundation, 33 W 42nd St, New York, NY 10036, USA. [5] Aix Marseille Univ, Université de Toulon, CNRS, CPT, Marseille 13009, France. [6] Dipartimento di Fisica ed Astronomia, Universitá di Catania and INFN, 95123 Catania, Italy. [7] Complexity Science Hub Vienna, Josefstädter Strasse 39, Vienna 1080, Austria. Correspondence and requests for materials should be addressed to V.L. (email: v.latora@qmul.ac.uk)

Complex networks describe well the connectivity of systems of various nature[1,2] and are widely used as the underlying —and possibly multilayered[3]—social structure on which dynamical processes[4,5], such as disease spreading[6], diffusion and adoption of innovation[7–9], and opinion formation[10] occur. For example, when modeling an epidemic spreading in a population[6], the transmission between infectious and healthy individuals is typically assumed: (i) to occur through pairwise interactions between infectious and healthy individuals, and (ii) to be caused even by a single exposure of a healthy individual to an infectious one. Such processes of simple contagion can be conveniently represented by transmission mechanisms along the links of the network of contacts between individuals.

When dealing instead with social contagion phenomena, such as the adoption of norms, behaviors or new products, or the diffusion of rumors or fads, the situation is more complex. Simple epidemic-like contagion can suffice to describe some cases, such as easily convincing rumors or domino effects[11]. In other situations, however, they do not provide a satisfactory description, especially in those cases where more complex dynamics of peer influence and reinforcement mechanisms are at work[12]. Complex contagion mechanisms have been proposed to account for these effects. As defined by Centola and Macy[11]: "a contagion is complex if its transmission requires an individual to have contact with two or more sources of activation", i.e. if a "contact with a single active neighbor is not enough to trigger adoption". Complex contagion can hence be broadly defined as a process in which exposure to multiple sources presenting the same stimulus is needed for the contagion to occur. Empirical evidence that contagion processes including multiple exposure can be needed to describe social contagion has been provided in various contexts and experiments[13–17].

Modeling of social contagion processes has been driven by these considerations in several directions. Threshold models assume that, in order to adopt a novel behavior, an individual needs to be convinced by a fraction of his/her social contacts larger than a given threshold[11,16,18–21]. The processes considered in such models are usually deterministic. Another modeling framework for social contagion relies instead on generalizations of epidemic-like processes, with stochastic contagion processes whose rates might depend on the number of sources of exposure to which an individual is linked to, i.e., with a complex contagion flavor[15,21–26]. All these models are however still defined on networks of interactions between individuals: even when multiple interactions are needed for a contagion to take place, in both threshold and epidemic-like models, the fundamental building blocks of the system are pairwise interactions, structurally represented by the links of the network on which the process is taking place.

Here, we propose to go further and take into account that contagion can occur in different ways, either through pairwise interactions (the links of a network) or through group interactions, i.e., through higher-order structures. Indeed, while an individual can be convinced independently by each of his/her neighbors (simple contagion), or by the successive exposure to the arguments of different neighbors (complex contagion), a fundamentally different mechanism is at work if the neighbors of an individual convince him/her in a group interaction. For example, we can adopt a new norm because of two-body processes, which means we can get convinced, separately, by each one of our first neighbors in our social network who have already adopted the norm. However, this is qualitatively different from a mechanism of contagion in which we get convinced because we are part of a social group of three individuals, and our two neighbors are both adopters. In this case the contagion is a three-body process, which mimics the simplest multiple source of

reinforcement that induces adoption. The same argument can easily be generalized to larger group sizes.

To build a modeling framework based on these ideas, we formalize a social group as a simplex, and we adopt simplicial complexes as the underlying structure of the social system under consideration (see Fig. 1a, b). This simplicial representations is indeed more suited than networks to describe the co-existence of pairwise and higher-order interactions. We recall that, in its most basic definition, a $k$-simplex $\sigma$ is a set of $k + 1$ vertices $\sigma = [p_0, \ldots, p_k]$. It is then easy to see the difference between a group interaction among three elements, which can be represented as a 2-simplex or "full" triangle $[p_0, p_1, p_2]$, and the collection of its edges, $[p_0, p_1]$, $[p_0, p_2]$, $[p_1, p_2]$. Just like a collection of edges defines a network, a collection of simplices defines a simplicial complex. Formally, a simplicial complex $\mathcal{K}$ on a given set of vertices $\mathcal{V}$, with $|\mathcal{V}| = N$, is a collection of simplices, with the extra requirement that if simplex $\sigma \in \mathcal{K}$, then all the subsimplices $\nu \subset \sigma$ built from subsets of $\sigma$ are also contained in $\mathcal{K}$. Such a requirement, which makes simplicial complexes a special type of hypergraphs (see Supplementary Note 4), seems appropriate in the definition of higher-dimensional groups in the context of social systems, and simplicial complexes have indeed been used to represent social aggregation in human communication[27]. Removing this extra requirement would imply, for instance, modeling a group interaction of three individuals without taking into account also the dyadic interactions among them. The same argument can be extended to interactions of four or more individuals: it is reasonable to assume that the existence of high-order interactions implies the presence of the lower-order interactions. For simplicity and coherence with the standard network nomenclature, we call nodes (or vertices) the 0-simplices and links (or edges) the 1-simplices of a simplicial complex $\mathcal{K}$, while 2-simplices correspond to the ("full") triangles, 3-simplices to the tetrahedra of $\mathcal{K}$, and so on (see Fig. 1a). Simplicial complexes, differently from networks, can thus efficiently characterize interactions between any number of units[28,29]. Simplicial complexes are not a new idea[30], but the interest in them has been renewed[29,31,32] thanks to the availability of new data sets and of recent advances in topological data analysis techniques[33]. In particular, they recently proved to be useful in describing the architecture of complex networks[34–36] functional[37–39] and structural brain networks[40], protein interactions[41], semantic networks[42], and co-authorship networks in science[43].

Here, we thus propose a new modeling framework for social contagion, namely a model of "simplicial contagion": this epidemic-like model of social contagion on simplicial complexes takes into account the fact that contagion processes occurring through a link or through a group interaction both exist and have different rates. Our model therefore combines stochastic processes of simple contagion (pairwise interactions) and of complex contagion occurring through group interactions in which an individual is simultaneously exposed to multiple sources of contagion. We perform extensive numerical simulations on both empirical data and synthetic simplicial complexes and develop as well an analytical approach in which we derive and solve the mean-field equations describing the evolution of density of infected nodes. We show both numerically and analytically that the higher-order interactions lead to the emergence of new phenomena, changing the nature of the transition at the epidemic threshold from continuous to discontinuous and leading to the appearance of a bistable region of the parameter space where both healthy and endemic asymptotic states co-exist. The mean-field analytical approach correctly predicts the steady-state dynamics, the position and the nature of the transition and the location of the bistable region. We also show that, in the bistable region, a critical mass is needed to reach the endemic state, reminding of

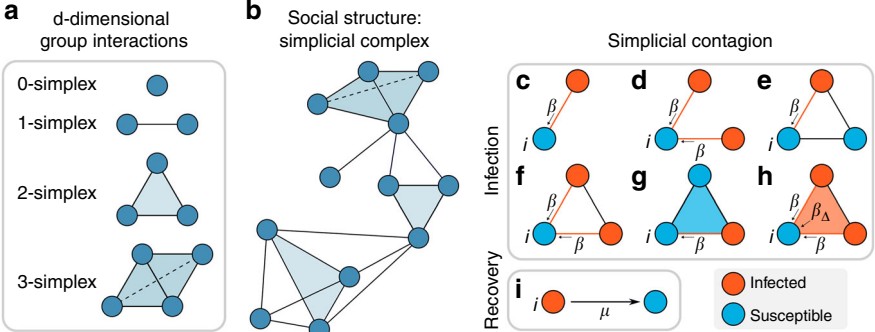

**Fig. 1** Simplicial contagion model (SCM). The underlying structure of a social system is made of simplices, representing d-dimensional group interactions (**a**), organized in a simplicial complex (**b**). **c–h** Different channels of infection for a susceptible node $i$ in the simplicial contagion model (SCM) of order $D = 2$. Susceptible and infected nodes are colored in blue and red, respectively. Node $i$ is in contact with one (**c**, **e**) or more (**d**, **f**) infected nodes through links (1-simplices), and it becomes infected with probability $\beta$ at each timestep through each of these links. **g**, **h** Node $i$ belongs to a 2-simplex (triangle). In **g** one of the nodes of the 2-simplex is not infected, so $i$ can only receive the infection from the (red) link, with probability $\beta$. In **h** the two other nodes of the 2-simplex are infected, so $i$ can get the infection from each of the two 1-faces (links) of the simplex with probability $\beta$, and also from the 2-face with probability $\beta_2 = \beta_\Delta$. **i** Infected nodes recover with probability $\mu$ at each timestep, as in the standard SIS model

the recently observed minimal size of committed minorities required to initiate social changes[44].

## Results

**The contagion model**. In order to model a simplicial contagion process, we associate a dynamical binary state variable $x$ to each of the $N$ vertices of $\mathcal{K}$, such that $x_i(t) \in \{0, 1\}$ represents the state of vertex $i$ at time $t$. Using a standard notation, we divide the population of individuals into two classes of susceptible (S) and infectious (I) nodes, corresponding respectively to the values 0 and 1 of the state variable $x$. In the context of adoption processes, the state $I$ represents individuals who have adopted a behavior. At each time $t$, the macroscopic order parameter is given by the density of infectious nodes $\rho(t) = \frac{1}{N}\sum_{i=1}^{N} x_i(t)$. The model we propose here, the so-called Simplicial Contagion Model (SCM) of order $D$, with $D \in [1, N-1]$, is governed by a set of $D$ control parameters $B = \{\beta_1, \beta_2, \ldots, \beta_D\}$, whose elements represent the probability per unit time for a susceptible node $i$ that participates to a simplex $\sigma$ of dimension $D$ to get the infection from each one of the subfaces composing $\sigma$, under the condition that all the other nodes of the subface are infectious. In practice, with this notation, $\beta_1$ is equal to the standard probability of infection $\beta$ that a susceptible node $i$ gets the infection from an infected neighbor $j$ through the link $(i, j)$ (corresponding to the process $S + I \rightarrow 2I$). Similarly, the second parameter $\beta_2 \equiv \beta_\Delta$ corresponds to the probability per unit time that node $i$ receives the infection from a "full" triangle (2-simplex) $(i, j, k)$ in which both $j$ and $k$ are infectious, $\beta_3 = \beta_\boxtimes$ from a group of size 4 (3-simplex) to which it belongs, and so on. Such processes can be represented as $\text{Simp}(S, nI) \rightarrow \text{Simp}((n+1)I)$: a susceptible node, part of a simplex of $n + 1$ nodes among which all other $n$ nodes are infectious, becomes infectious with probability per unit time $\beta_n$. Thanks to the simplicial complex requirements that all subsimplices of a simplex are included, contagion processes in a $n$-simplex among which $p < n$ nodes are infectious are also automatically considered, each of the $n + 1 - p$ susceptible nodes being in a simplex of size $p + 1$ with the $p$ infectious ones. Notice, however, that this assumption can be dropped and the contagion model extended to the case of hypergraphs[45,46] (see Supplementary Note 4). Figure 1c–h illustrates the concrete example of the six possible ways in which a susceptible node $i$ can undergo social contagion for an SCM of order $D = 2$ with parameters $\beta$ and $\beta_\Delta$. Finally, the recovery dynamics ($I \rightarrow S$) is controlled by the node-independent recovery probability $\mu$ (Fig. 1i). Notice that the SCM of order $D$ reduces to the standard SIS model on a network when $D = 1$, since in this case the infection can only be transmitted through the links of $\mathcal{K}$.

**Simplicial contagion on real-world simplicial complexes**. To explore the phenomenology of the simplicial contagion model, we first consider its evolution on empirical social structures. To this aim, we consider publicly available data sets describing face-to-face interactions collected by the SocioPatterns collaboration[47]. Face-to-face interactions represent indeed a typical example in which group encounters are fundamentally different from sets of binary interactions and can naturally be encoded as simplices. The time-resolved nature of the data allows us to create simplicial complexes describing the aggregated social structure, as described in Methods. For simplicity, we only consider simplices of dimension up to $D = 2$. We consider data on interactions collected in four different social contexts: a workplace, a conference, a hospital and a high school (see Methods for details on the data sets).

We simulate the SCM over the simplicial complexes obtained from the four data sets as described in Methods. In particular, we start with an initial density $\rho_0$ of infectious nodes and we run the simulations by taking into consideration all the possible channels of infection illustrated in Fig. 1c–h. We stop a simulation if an absorbing state is reached, otherwise we compute the average stationary density of infectious nodes $\rho^\star$ by averaging the values measured in the last 100 time-steps after reaching a stationary state. The results are averaged over 120 runs obtained with randomly placed initial infectious nodes with the same density $\rho_0$. Moreover, the different data sets correspond to different densities of 1- and 2-simplices (see Supplementary Note 1). We thus rescale the infectivity parameters $\beta$ and $\beta_\Delta$ respectively by the average degree $\langle k \rangle$ and by the average number of 2-simplices incident on a node, $\langle k_\Delta \rangle$. We finally express all results as functions of the rescaled parameters $\lambda = \beta\langle k \rangle/\mu$ and $\lambda_\Delta = \beta_\Delta\langle k_\Delta \rangle/\mu$.

Figure 2 shows the resulting prevalence curves for the four data sets (see also Supplementary Note 5). In each panel (Fig. 2b, d, f, h), the average fraction of infected nodes $\rho^\star$ in the stationary state is plotted as a function of the rescaled infectivity $\lambda = \beta\langle k \rangle/\mu$ for simulations of the SCM with $\lambda_\Delta = 0.8$ (black triangles) and $\lambda_\Delta = 2$ (orange squares). For comparison, we also plot the case $\lambda_\Delta = 0$, which is equivalent to the standard SIS model with no higher-order effects (blue circles). We observe two radically different

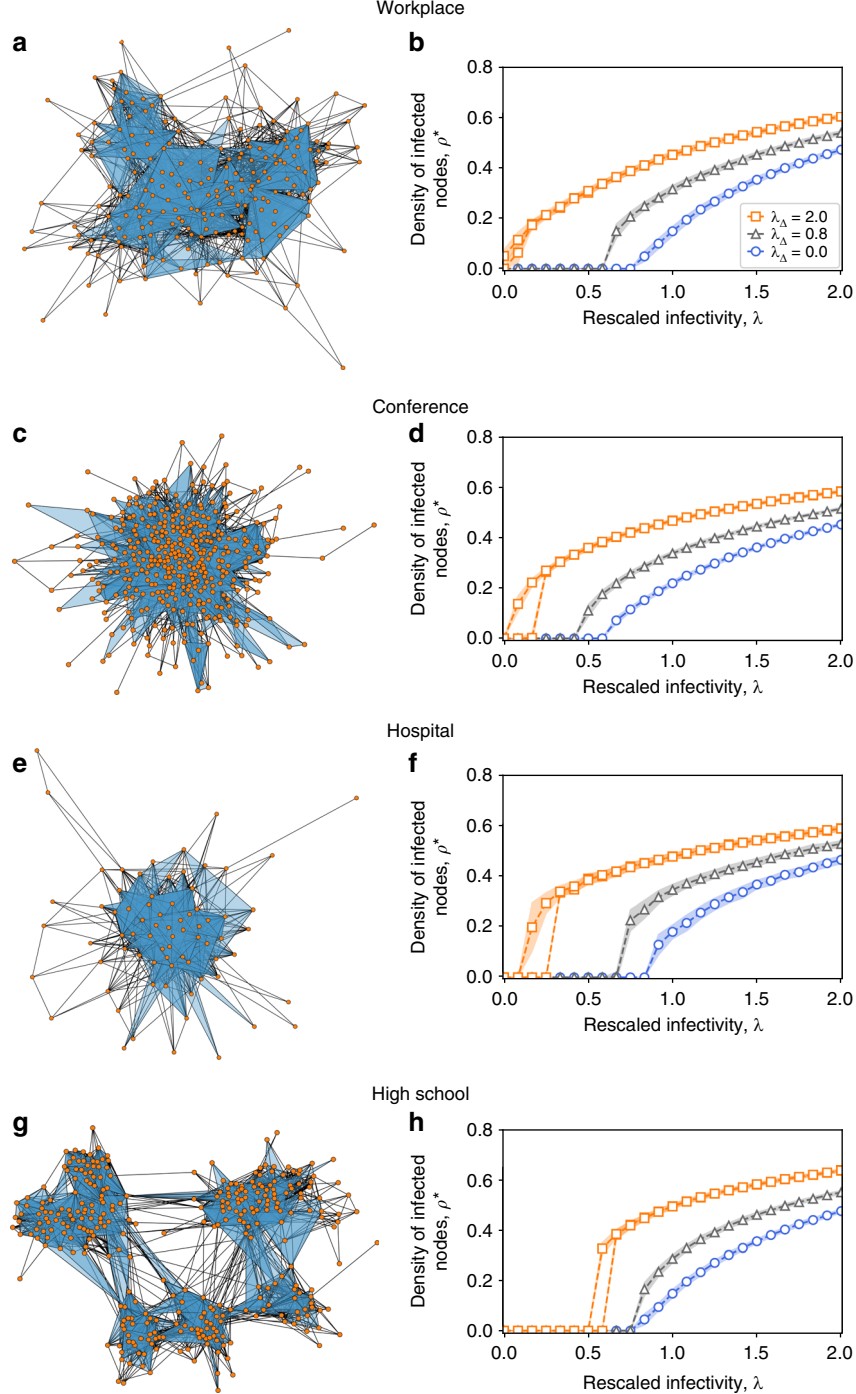

**Fig. 2** SCM of order $D = 2$ on real-world higher-order social structures. Simplicial complexes are constructed from high-resolution face-to-face contact data recorded in four different context: **a** a workplace, **c** a conference, **e** a hospital and **g** a high school. Prevalence curves are respectively reported in panels **b**, **d**, **f** and **h**, in which the average fraction of infectious nodes obtained in the numerical simulations is plotted against the rescaled infectivity $\lambda = \beta\langle k\rangle/\mu$ for different values of the rescaled parameter $\lambda_\Delta = \beta_\Delta\langle k_\Delta\rangle/\mu$, namely $\lambda_\Delta = 0.8$ (black triangles) and $\lambda_\Delta = 2$ (orange squares). The blue circles denote the simulated curve for the equivalent standard SIS model ($\lambda_\Delta = 0$), which does not consider higher-order effects. For $\lambda_\Delta = 2$ a bistable region appears, where healthy and endemic states co-exist

behaviors for the two values of $\lambda_\Delta \neq 0$. For $\lambda_\Delta = 0.8$, the density of infectious nodes varies as a function of $\lambda$ in a very similar way to the case $\lambda_\Delta = 0$ (simple contagion), with a continuous transition. For $\lambda_\Delta = 2$ we observe instead the appearance of an endemic state with $\rho^\star > 0$ at a value of $\lambda^c$ well below the epidemic threshold of the other two cases. Furthermore, this transition appears to be discontinuous, and an hysteresis loop appears in a bistable region,

where both healthy $\rho^\star = 0$ and endemic $\rho^\star > 0$ states can co-exist (dashed orange lines): in this parameter region, the final state depends on the initial density of infectious nodes $\rho_0$.

The simplicial complexes used in these simulations correspond to various social contexts and different densities of 1- and 2-simplices, and yield a similar phenomenology. These empirical structures however exhibit distributions of generalized degrees that

are not well peaked around their average values (see Supplementary Note 1), and do not allow us to systematically explore size effects. To better understand the phenomenology of the simplicial contagion model, we thus now explore its behavior on synthetic simplicial complexes with controlled properties.

**Simplicial contagion on synthetic simplicial complexes.** A range of models for random simplicial complexes have been proposed so far, starting from the exponential random simplicial complex, the growing and generalized canonical ensemble[48–50] and the simplicial configuration models[51] to the simplicial activity-driven model[52] generalizing the activity-driven temporal network model[53]. While these yield Erdös−Rényi-like models[54,55] of arbitrary complexity, here we are interested in models generating simplicial complexes with simplices of different dimension in which we can control and tune the expected local connectivity, e.g. the number of edges and "full" triangles a node belongs to. We therefore propose a new model to construct random simplicial complexes, the RSC model, which allows us to maintain the average degree of the nodes, $\langle k_1 \rangle$, fixed, while varying at the same time the expected number of "full" triangles (2-simplices) $\langle k_\Delta \rangle$ incident on a node. The RSC model of dimension $D$ has $D + 1$ parameters, namely the number of vertices $N$ and $D$ probabilities $\{p_1, …, p_k, …, p_D\}$, $p_k \in [0, 1]$, which control for the creation of $k$-simplices up to dimension $D$. For the purpose of this study we limit the RSC model to $D = 2$, which restricts the set of required parameters to $(N, p_1, p_2)$, but the procedure could easily be extended to larger $D$. The model works as follows. We first create 1-simplices (links) as in the Erdös −Rényi model[56], by connecting any pair $(i, j)$ of vertices with probability $p_1$. Similarly, 2-simplices are then created by connecting any triplet $(i, j, k)$ of vertices with probability $p_2 \equiv p_\Delta$. Notice that simplicial complexes built in this way are radically different from the clique complexes obtained from Erdös−Rényi graphs[54], in which every subset of nodes forming a clique is automatically "promoted" to a simplex. Contrarily, in a simplicial complex generated by the RSC model proposed here, a 2-simplex $(i, j, k)$ does not come from the promotion of an "empty" triangle composed by three 1-simplices $(i, j)$, $(j, k)$, $(k, i)$ to a "full triangle" $(i, j, k)$. This also means that the model allows for the presence of $(k + 1)$-cliques that are not considered $k$-simplices; therefore, it is able to generate simplicial complexes having both "empty" and "full" triangles, respectively encoding three 2-body interactions and one 3-body interactions (as for instance in Fig. 1b). The expected average numbers of 1- and 2-simplices incident on a node, noted $\langle k \rangle$ and $\langle k_\Delta \rangle$, are easy to calculate (see Methods). Therefore, for any given size $N$, we can produce simplicial complexes having desired values of $\langle k \rangle$ and $\langle k_\Delta \rangle$ by appropriately tuning $p_1$ and $p_\Delta$. More details about the construction of the model and the tuning of the parameters are provided in the "Methods" section, while the agreement between the expected values of $\langle k \rangle$ and $\langle k_\Delta \rangle$ with the empirical averages obtained from different realizations of the model is discussed in Supplementary Note 1.

We simulate the SCM over an RSC created with the procedure described above, with $N = 2000$ nodes, $\langle k \rangle \simeq 20$ and $\langle k_\Delta \rangle \simeq 6$. As for the real-world simplicial complexes, we start with a seed of $\rho_0$ infectious nodes placed at random and we compute the average stationary density of infectious $\rho^\star$ by averaging over different runs, each one using a different instance of the RSC model. Results are shown in Fig. 3a, where the average fraction of infected nodes, as obtained by the simulations, is plotted as a function of the rescaled infectivity $\lambda = \beta \langle k \rangle$ for a ($D = 2$) SCM with $\lambda_\Delta = 0.8$ (white squares), $\lambda_\Delta = 2.5$ (filled blue circles) and $\lambda_\Delta = 0$ (light blue circles).

Despite the very different properties of the underlying structure, the dynamics of the SCM on the RSC is very similar to the one observed on the real-world simplicial complexes. For $\lambda_\Delta = 0.8$ the model behaves similarly to a simple contagion model ($\lambda_\Delta = 0$), with a continuous transition at $\lambda^c = 1$, the well-know epidemic threshold of the standard SIS model on homogeneous networks. When a higher value of $\lambda_\Delta$ is considered ($\lambda_\Delta = 2.5$), the epidemic can be sustained below $\lambda^c = 1$, and both an epidemic-free and an endemic state are present in the region $\lambda^c < \lambda < 1$, with appearance of a hysteresis loop (see the filled blue circles in Fig. 3a). In this region, we obtain $\rho(t \to \infty) = 0$ for $\rho(t = 0) = 0.01$, while $\rho(t \to \infty) > 0$ for $\rho(t = 0) = 0.4$. The size-dependence of the hysteresis loop is shown in Supplementary Note 2 to be very small. The dependency from the initial conditions is also further illustrated in Fig. 3b, in which the temporal dynamics of single runs are shown. The various curves show how the density of infected nodes $\rho(t)$ evolves when initial seeds of infected nodes of different sizes are considered. Each color corresponds to a different value of $\rho_0$, with brighter colors representing higher initial densities of infected individuals. The figure clearly shows the presence of a threshold value for $\rho_0$, such that $\rho(t)$ goes to the absorbing state $\rho(t) = 0$ if $\rho_0$ is smaller than the threshold, and to a nontrivial steady state if the initial density is above the threshold.

**Mean field approach.** In order to study more extensively this phenomenology as $\lambda_\Delta$ and $\lambda$ vary, and to further characterize the discontinuous transition, we consider a mean field (MF) description of the SCM, under a homogeneous mixing hypothesis[57]. Given the set of infection probabilities $B \equiv \{\beta_\omega,\ \omega = 1, …, D\}$ and a recovery probability $\mu$, we assume the independence between the states $x_i(t)$ and $x_j(t)$ $\forall i, j \in \mathcal{V}$, and we write an MF expression for the temporal evolution of the density of infected nodes $\rho(t)$ as:

$$d_t \rho(t) = -\mu \rho(t) + \sum_{\omega=1}^{D} \beta_\omega \langle k_\omega \rangle \rho^\omega(t)[1 - \rho(t)], \quad (1)$$

where, for each $\omega = 1, \cdots, D$, $k_\omega(i) = k_{\omega,0}(i)$ is the generalized (simplicial) degree of a 0-dimensional face (node $i$), i.e., the number of $\omega$-dimensional simplices incident to the node $i$[49,50], and $\langle k_\omega \rangle$ is its average over all the nodes $i \in \mathcal{V}$. With this approximation we assume that the local connectivity of the nodes is well described by globally averaged properties, such as the average generalized degree. We can immediately check that in the case $D = 1$ we recover the standard MF equation for the SIS model, which leads to the well-known stationary state solutions $\rho_1^{*[D=1]} = 0$ and $\rho_2^{*[D=1]} = 1 - \mu/(\beta \langle k \rangle)$. The absorbing state $\rho_1^{*[D=1]} = 0$ is the only solution for $\beta \langle k \rangle / \mu < 1$, i.e., below the epidemic threshold. When $\beta \langle k \rangle / \mu > 1$, this state becomes unstable while the solution $\rho_2^{*[D=1]}$ becomes a stable fixed point of the dynamics. The transition between these two regimes is continuous at $\beta \langle k \rangle / \mu = 1$.

Let us now focus on a more interesting but still analytically tractable case in which we extend the contagion dynamics up to dimension $D = 2$, so that Eq. (1) reads:

$$d_t \rho(t) = -\mu \rho(t) + \beta \langle k \rangle \rho(t)[1 - \rho(t)] + \beta_\Delta \langle k_\Delta \rangle \rho^2(t)[1 - \rho(t)], \quad (2)$$

where $\langle k_\Delta \rangle \equiv \langle k_2 \rangle$. By defining as before $\lambda = \beta \langle k \rangle / \mu$ and $\lambda_\Delta = \beta_\Delta \langle k_\Delta \rangle / \mu$, and by rescaling the time by $\mu$, we can rewrite eq. (2) as:

$$d_t \rho(t) = -\rho(t)\big(\rho(t) - \rho_{2+}^*\big)\big(\rho(t) - \rho_{2-}^*\big), \quad (3)$$

where $\rho_{2+}^*$ and $\rho_{2-}^*$ are the solutions of the second-order equation

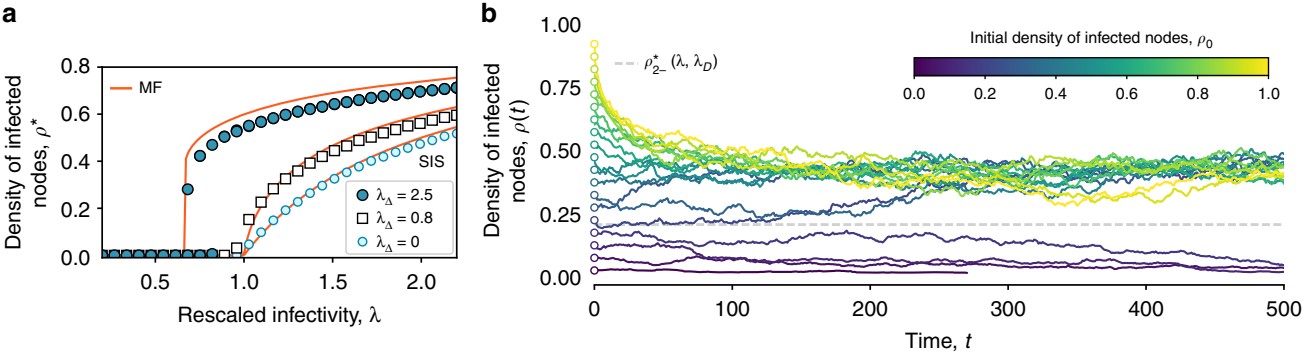

**Fig. 3** SCM of order $D = 2$ on a synthetic random simplicial complex (RSC). The RSC is generated with the procedure described in this manuscript, with parameters $N = 2000$, $p_1$ and $p_\Delta$ tuned in order to produce a simplicial complex with $\langle k \rangle \sim 20$ and $\langle k_\Delta \rangle \sim 6$. **a** The average fraction of infected obtained by means of numerical simulations is plotted against the rescaled infectivity $\lambda = \beta\langle k \rangle/\mu$ for $\lambda_\Delta = 0.8$ (white squares) and $\lambda_\Delta = 2.5$ (filled blue circles). The light blue circles give the numerical results for the standard SIS model ($\lambda_\Delta = 0$) that does not consider higher-order effects. The red lines correspond to the analytical mean field solution described by Eq. (3). For $\lambda_\Delta = 2.5$ we observe a discontinuous transition with the formation of a bistable region where healthy and endemic states co-exist. **b** Effect of the initial density of infected nodes, shown by the temporal evolution of the densities of infectious nodes (a single realization is shown for each value of the initial density). The infectivity parameters are set within the range in which we observe a bistable region ($\lambda = \beta \langle k \rangle/\mu = 0.75$, $\lambda_\Delta = \beta_\Delta\langle k_\Delta \rangle/\mu = 2.5$). Different curves—and different colors—correspond to different values for the initial density of infectious nodes $\rho_0 \equiv \rho$ (0). The dashed horizontal line corresponds to the unstable branch $\rho^*_{2-}$ of the mean field solution given by Eq. 4, which separates the two basins of attraction

$1 - \lambda(1 - \rho) - \lambda_\Delta\rho(1 - \rho) = 0$. We thus obtain:

$$\rho^*_{2\pm} = \frac{\lambda_\Delta - \lambda \pm \sqrt{(\lambda - \lambda_\Delta)^2 - 4\lambda_\Delta(1 - \lambda)}}{2\lambda_\Delta}. \quad (4)$$

The steady-state equation $d_t\rho(t) = 0$ has thus up to three solutions in the acceptable range $\rho \in [0, 1]$. The solution $\rho^*_1 = 0$ corresponds to the usual absorbing epidemic-free state, in which all the individuals recover and the spreading dies out. A careful analysis of the stability of this state and of the two other solutions $\rho^*_{2+}$ and $\rho^*_{2-}$ is however needed to fully characterize the phase diagram of the system.

Let us first consider the case $\lambda_\Delta \le 1$. It is possible to show that $\rho^*_{2-}$, when it is real-valued, is always negative, i.e., it is not an acceptable solution. Moreover, $\rho^*_{2+}$ is positive for $\lambda > 1$ and negative for $\lambda < 1$. In the regime $\lambda_\Delta \le 1$ therefore, if $\lambda < 1$, the only acceptable solution to $d_t\rho(t) = 0$ is $\rho^*_1 = 0$; contrarily, for $\lambda > 1$, since $\rho^*_{2-} < 0$ and $\rho^*_{2+} > 0$, Eq. (3) shows that $d_t\rho(t)$ is positive at small $\rho(t)$: the absorbing state $\rho^*_1 = 0$ is thus unstable and the solution $\rho^*_{2+}$ is stable. As $\rho^*_{2+} = 0$ for $\lambda = 1$, the transition at the epidemic threshold $\lambda = 1$ is continuous. In conclusion, when $\lambda_\Delta \le 1$, the transition is similar to the one of the standard SIS model with $\lambda_\Delta = 0$.

Let us now consider the case of $\lambda_\Delta > 1$. Then, for $\lambda < \lambda^c = 2\sqrt{\lambda_\Delta} - \lambda_\Delta$, both $\rho^*_{2+}$ and $\rho^*_{2-}$ are outside the real domain, and the only steady state is the absorbing one $\rho^*_1 = 0$. Note that $\lambda^c < 1$, since $\lambda_\Delta > 1$. For $\lambda > \lambda^c$, we thus have two possibilities to consider. If $\lambda > 1$, we can show that $\rho^*_{2-} < 0 < \rho^*_{2+}$. Equation (3) shows then that, for small $\rho(t)$, $d_t\rho(t) > 0$: as above, the absorbing state $\rho^*_1 = 0$ is unstable and the density of infectious nodes tends to $\rho^*_{2+}$ in the large time limit; if instead $\lambda^c < \lambda < 1$, we obtain that $0 < \rho^*_{2-} < \rho^*_{2+}$. Then, still from Eq. (3), we obtain that $d_t\rho(t) < 0$ for $\rho(t)$ between 0 and $\rho^*_{2-}$, and that $d_t\rho(t) > 0$ for $\rho(t)$ between $\rho^*_{2-}$ and $\rho^*_{2+}$. As a result, both $\rho^*_1 = 0$ and $\rho^*_{2+}$ are stable steady states of the dynamics, while $\rho^*_{2-}$ is an unstable solution. Most interestingly, the long time limit of the dynamics depends then on the initial conditions. Indeed, if the initial density of infectious nodes, $\rho(t = 0)$, is below $\rho^*_{2-}$, the short time derivative of $\rho(t)$ is negative, so that the density of infectious nodes decreases and the system tends to the absorbing state: $\rho(t) \underset{t\to\infty}{\longrightarrow} 0$. On the other hand, if the initial density $\rho(t = 0)$ is

large enough (namely, larger than $\rho^*_{2-}$), the dynamical evolution equation (3) pushes the density towards the value $\rho^*_{2+}$, i.e. $\rho(t) \underset{t\to\infty}{\longrightarrow} \rho^*_{2+}$. Since $\rho^*_{2+} > 0$, the transition at $\lambda_c$ is discontinuous.

We illustrate these results by showing in Fig. 4a the solutions $\rho^*_1$, $\rho^*_{2+}$ and $\rho^*_{2-}$ as a function of $\lambda$ and for different values of $\lambda_\Delta$. The vertical line corresponds to the standard epidemic threshold for the SIS model ($\lambda_\Delta = 0$). Dashed lines depict unstable branches, as given by $\rho^*_{2-}$. We emphasize again two important points. First, for $\lambda_\Delta > 1$ we observe a discontinuous transition at $\lambda^c = 2\sqrt{\lambda_\Delta} - \lambda_\Delta$, instead of the usual continuous transition at the epidemic threshold. Second, for $\lambda^c < \lambda < 1$ the final state depends on the initial density of infectious nodes, as described above: the absorbing state $\rho^*_1 = 0$ is reached if the initial density $\rho$ $(t = 0)$ is below the unstable steady-state value $\rho^*_{2-}$; on the contrary, if $\rho(t = 0)$ is above this value, the system tends to a finite density of infectious nodes equal to $\rho^*_{2+}$. In other words, a critical mass is needed to reach the endemic state, reminding of the recently observed minimal size of committed minorities required to initiate social changes[44].

Figure 4b is a two-dimensional phase diagram showing $\rho^*_{2+}$ for different values of $\lambda$ and $\lambda_\Delta$. Lighter colors correspond to higher values of the stationary density of infectious nodes, while the dashed vertical line corresponds to the epidemic threshold of the standard (without higher-order effects) SIS model, namely $\lambda = 1$. For $\lambda_\Delta \le 1$ (below the dashed horizontal line) the transition as $\lambda$ crosses 1 is seen to be continuous, while, for $\lambda_\Delta > 1$, the transition is clearly discontinuous along the curve $\lambda^c = 2\sqrt{\lambda_\Delta} - \lambda_\Delta$ (dash-dotted line). The analytical values of $\rho^*_{2+}$ are also reported as continuous red lines in Fig. 3a and compared to the results of the simulations, showing in this way the accuracy of the mean field approach just described. In addition, Fig. 3b shows that the unstable solution $\rho^*_{2-}$ accurately separates the two basins of attractions for the dynamics, i.e., it defines the critical initial density of infected $\rho_0$ that determines whether the long-term dynamics reaches the healthy state or the endemic one. Notice that the mean field approach is in fact able to correctly capture both the position of the thresholds and the discontinuous nature of the transition for the SCM with $\lambda_\Delta > 1$.

We finally note that, while a general solution for general $D$ with arbitrary parameters $\{\beta_\omega\}$ remains out of reach, it is possible to

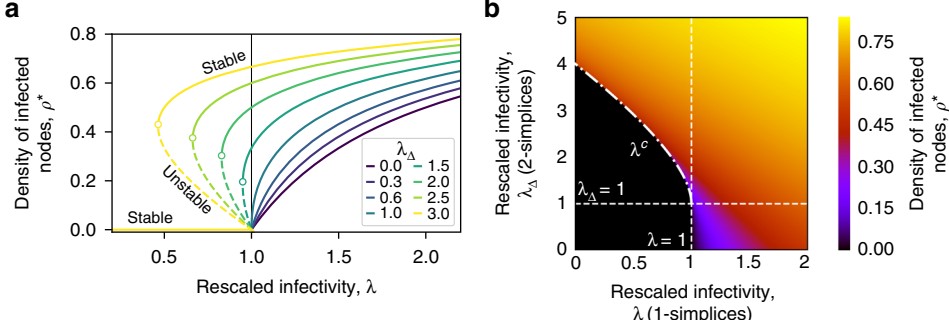

**Fig. 4** Phase diagram of the SCM of order $D = 2$ in mean field approximation. **a** The stationary solutions $\rho^\star$ given by Eq. (4) are plotted as a function of the rescaled link infectivity $\lambda = \beta \langle k \rangle / \mu$. Different curves correspond to different values of the triangle infectivity $\lambda_\Delta = \beta_\Delta \langle k_\Delta \rangle / \mu$. Continuous and dashed lines correspond to stable and unstable branches respectively, while the vertical line denotes the epidemic threshold $\lambda^c = 1$ in the standard SIS model that does not consider higher-order effects. For $\lambda_\Delta \leq 1$ the higher-order interactions only contribute to an increase in the density of infected individuals in the endemic state, while they leave the threshold unchanged. Conversely, when $\lambda_\Delta > 1$ we observe a shift of the epidemic threshold, and a change in the type of transition, which becomes discontinuous. **b** Heatmap of the stationary solution $\rho^\star$ given by Eq. (4) as a function of the rescaled infectivities $\lambda = \beta \langle k \rangle / \mu$ and $\lambda_\Delta = \beta_\Delta \langle k_\Delta \rangle / \mu$. The black area corresponds to the values of $(\lambda, \lambda_\Delta)$ such that the only stable solution is $\rho_1^* = 0$. The dashed vertical line corresponds to $\lambda = 1$, the epidemic threshold of the standard SIS model without higher-order effects. The dash-dotted line represents the points $(\lambda^c, \lambda_\Delta)$, with $\lambda^c = 2\sqrt{\lambda_\Delta} - \lambda_\Delta$, where the system undergoes a discontinuous transition

show that the phenomenology obtained for $D = 2$ is also observed for specific cases with $D \geq 3$. We consider indeed in the Supplementary Note 3 two cases: $D = 3$ with $\beta_2 = 0$ and general $D > 3$ with $\beta_1 = \cdots = \beta_{D-1} = 0$. In both cases, we show the appearance of a discontinuous transition in the regime where the simple contagion $\beta_1$ is below threshold (i.e., $\beta_1 \langle k \rangle < \mu$): similarly to the case $D = 2$, this transition occurs as $\beta_D$, which describes the rate of the high-order contagion process, increases.

## Discussion

In summary, the simplicial model of contagion introduced in this work is able to capture the basic mechanisms and effects of higher-order interactions in social contagion processes. Our analytical results were derived in a mean field approximation and indeed quantatively compared to the nondescript simplicial complexes obtained in our random simplicial complex model (akin to ER simplicial complexes[55]). However, the framework we introduced is very general and the phenomenology robust, as seen from the results obtained on empirical data sets. It would be interesting to investigate the SCM on more general simplicial complexes with for instance heterogeneous generalized degree distribution or with community structures, and to consider simplicial complexes with emergent properties such as hyperbolic geometry[58–60], or temporally evolving simplicial complexes[52]. Furthermore, given that the SCM can be mapped on a model with hypergraphs if the hyperedges of different types are carefully chosen, it would be interesting to study the behavior of complex contagion processes on more general classes of hypergraphs[61,62]. Finally, we hope that the idea will be extended from spreading processes to other dynamical systems, for instance to Kuramoto-like models with higher-order terms. Developing and studying such systems might allow to better take into account higher-order dynamical effects in real data-driven models.

## Methods

**Data description and processing**. We consider four data sets of face-to-face interactions collected in different social contexts: a workplace (InVS15)[63], a conference (SFHH)[64], a hospital (LH10)[65] and a high school (Thiers13)[66]. In each case face-to-face interactions have been measured with a temporal resolution of 20 s. We first aggregated the data by using a temporal window of $\Delta t = 5$ min, and computed all the maximal cliques that appear. Since we limit our study to the case $D = 2$, we need to produce a clique complex formed by 1- and 2-simplices. Therefore, we considered all the 2- and 3-cliques and weight them according to

### Table 1 Statistics of real-world simplicial complexes

| Data set | Context | $\langle k \rangle$ | $\langle k_\Delta \rangle$ | $\langle k \rangle^{\text{aug}}$ | $\langle k_\Delta \rangle^{\text{aug}}$ |
|---|---|---|---|---|---|
| InVS15 | Workplace | 16.9 | 7.0 | 21.0 | 7.0 |
| SFHH | Conference | 15.0 | 7.6 | 21.6 | 7.7 |
| LH10 | Hospital | 19.1 | 17.1 | 25.7 | 17.5 |
| Thiers13 | High school | 20.1 | 10.9 | 32.0 | 11.1 |

Average generalized degree of the four real-world simplicial complexes constructed from the considered data sets (before and after the data augmentation)

their frequency. Note that while higher-dimensional cliques are not included in the final simplicial complex, their sub-cliques up to size 3 are considered in the counting. We then retained 20% of the simplices with the largest number of appearances. The thresholded simplicial complexes obtained in this way are those used in Supplementary Fig. 6. Their connectivity properties are summarized in Table 1.

To reduce finite size effects, we augmented the thresholded simplicial complexes as follows: for each data set we extracted the list of sizes of the maximal simplices, also called facets, and the list of pure simplicial degrees of nodes. We then duplicated these lists five times and used the extended lists as input for the simplicial configuration model, described in ref. [51]. The outputs of this procedure are simplicial complexes with the same statistical properties as the input complex but of significantly larger size. We used these augmented complexes as substrates for the simulations shown in Fig. 2.

**Construction of random simplicial complexes**. The random simplicial complex (RSC) model produces simplicial complexes of dimension $D = 2$ as follows. Given a set $\mathcal{V}$ of $N$ vertices we connect any two nodes $i, j \in \mathcal{V}$ with probability $p_1 \in [0, 1]$, so that the average degree, at this stage, is $(N - 1)p_1$. Then, for any $i, j, k \in \mathcal{V}$, we add a 2-simplex $(i, j, k)$ with probability $p_\Delta \in [0, 1]$. At this point each node has an average number $\langle k_\Delta \rangle = (N - 1)(N - 2)p_\Delta / 2$ of incident 2-simplices that also contribute to increase the degree of the nodes. The exact contribution can be calculated by considering the different scenarios in which a 2-simplex $(i, j, k)$ can be attached to a node $i$ already having some links due to the first phase of the RSC construction. More precisely, the degree $k_i$ of node $i$ is incremented by 2 for each 2-simplex $(i, j, k)$ such that neither the link $(i, j)$ nor the link $(i, k)$ are already present; this happens with probability $(1 - p_1)^2$. Analogously, if either the link $(i, j)$ is already present but not $(i, k)$, or vice-versa, the addition of the 2-simplex $(i, j, k)$ increases the degree of $i$ by 1. Since each case happens with the same probability $p_1(1 - p_1)$ the contribution is therefore $2p_1(1 - p_1)$. Overall, the degree $k_i$ increases on average by $2(1 - p_1)$ for each 2-simplex attached to $i$. Finally, for $p_1, p_\Delta \ll 1$, we can thus write the expected average degree $\langle k \rangle$ as the sum of the two contributions coming from the links and the 2-simplices, namely $\langle k \rangle \approx (N - 1)p_1 + 2\langle k_\Delta \rangle (1 - p_1)$. For any given size $N$, we can thus produce simplicial complexes having desired values of $\langle k \rangle$ and $\langle k_\Delta \rangle$ by fixing $p_1$ and $p_\Delta$ as:

$$p_1 = \frac{\langle k \rangle - 2\langle k_\Delta \rangle}{(N - 1) - 2\langle k_\Delta \rangle}, \tag{5}$$

$$p_\Delta = \frac{2\langle k_\Delta \rangle}{(N-1)(N-2)}. \tag{6}$$

## Data availability

The SocioPatterns data sets were downloaded from https://www.sociopatterns.org/datasets.

## Code availability

The code and data sets are available at: https://github.com/iaciac/simplagion.

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

## Acknowledgements

I.I. and V.L. acknowledge support from EPSRC Grant EP/N013492/1. I.I. acknowledges support from The Alan Turing Institute under the EPSRC Grant No. EP/N510129/1. G.P. acknowledges support from ADnD Grant by Compagnia San Paolo and from Intesa Sanpaolo Innovation Center. The funders had no role in study design, data collection and analysis, decision to publish, or preparation of the manuscript.

## Author contributions

I.I., G.P., A.B. and V.L. designed the study. I.I. and G.P. performed the numerical analysis. I.I., G.P., A.B. and V.L. wrote the paper.

## Additional information

**Competing interests:** The authors declare no competing interests.

