## [Peer Review File · Nature Communications]

Reviewers' comments:

Reviewer #1 (Remarks to the Author):

I have carefully read the manuscript Iacopini and others entitled as "Simplicial models of social contagion", which was submitted to Nature Communications for publications. The paper propose a new model and its solution to consider higher order structural correlations in the dynamics of social contagion phenomena. The authors, being well known experts in the field of simplicial complexes, provide a concept to consider not only dyadic social influence but group influence as well in models of social spreading by introducing higher-order structures using k-simplices in the network description. After introduction, the authors successfully demonstrate the numerical simulations of their model and its mean-field approximate solution. They show that for increasing higher-order patterns the spreading process reaches a larger density of nodes in its stationary state via a continuous phase-transition, while for larger density of 2-simplices it goes through a discontinuous phase-transition with a bistable region.

The paper is well written, addresses an important extension of well known modelling methods, and explains a new phenomena. On the other hand I have a few concerns and comments on the manuscript regarding its model choice, generalisation, while I also argue that some of the claims of the authors, e.g. the seed size dependence of the spreading process, is not evidently supported by the results, thus requires further explanation. Please find my comments below.

Major comments:

- In the abstract the authors mention that their contributions "are a first step to understand the role of high-order interactions in complex systems". Considering higher-order correlations in complex systems, especially in complex networks is not entirely new as even some of the authors have extensive published work in this area. Thus, claiming that this manuscript is the first step is certainly an overshoot.

- In the introduction the authors argue the difference between simple and complex contagion assigning the latter one as the model adequate for the modelling of social contagion. Complex contagion is modelled by threshold driven mechanisms [see definition by Centola&Macy 2007], which capture accumulated social influence what individuals mount towards their threshold of adoption. Complex contagion processes are deterministic in their simplest definition and adoption is not driven by any probabilistic rate of transmission. Contrary to this picture, the authors model social contagion with an SIS type of process, where transition between node states are stochastic and driven by probabilistic rates. The model used in this manuscript actually extends the SIS model with group infection, which is yet driven by a probabilistic rate and recovery and has nothing to do with thresholds and complex contagion. SIS model is a simple contagion process, meant originally to describe biological epidemics, and certainly may have some relevance in social contagion. So I would suggest for the authors to make this point clear in the introduction otherwise they blur their contributions between these two modelling paradigms.

- On page 4 the authors introduce the concept of a k-simplex as "We recall that a k-simplex σ is in its simplest definition a collection of $k-1$ vertices $\sigma = [p_0, \dots, p_{k-1}]$ ". In my understanding a k-simplex is the collection of $k+1$ vertices. This is also compatible with their examples on page 5 where they call 0-simplex a node, 1-simplex an edge, and 2-simplex a triangle.

- Characterising social groups with higher order simplices might be a too rigid approach due to the restrictive geometric definition of a simplex. This way I understand why the authors only considered 1 and 2 simplices as building blocks of a social network. On the other hand, getting away from the social concept (which I understood might be out of the scope of the paper) it would very interesting to see how the critical behaviour of an SIS process generalise for higher D values.

- On page 8, in the approximate definition of the average degree of their model network does not consider links which simultaneously appear via both link addition process (by adding random links and triangles). This may set off their MF approximation results, as the networks they consider has relatively high degree, thus the contribution of multiple links might not be negligible. The subtraction of double counted links might be easy and may improve the fit between numerical simulations and the mean-field curves e.g. on Fig.2.

- It would be interesting to see how the histeresis and the phase-transitions in Fig.2 depends on the system size.

- Page 12 and 13: The authors write that "the final state depends on the initial density of infectious for $\lambda_c < \lambda < 1$, that is, a critical mass is needed to reach the endemic state". First of all "infectious" of what? Second of all, I cannot find any evidence in their results suggesting this conclusion. It would be necessary to explain this more in details by referring precisely which details of which figures support this claim, or remove it from the manuscript and also from the abstract.

- In the conclusion the authors mention in the last sentence that their modelling might be important to distinguish between higher-order dynamic effects in real data. They should discuss this more in details, maybe giving examples how their method can be applied on data. Without this, the last sentence is irrelevant. Actually, one of the main shortcoming is a missing data-driven simulation study on a small real social networks with a simulated SIS process.

Minor comments:

- Page 7: Typo at "in which both j and k are infectious, ,"

- On page 9 and later, calling a phase-transition explosive suggest a certain mechanisms (present in explosive percolations), which is not characterising the actual system at all. I would suggest to call it discontinuous or 1st order phase-transition.

- Page 11, 1st paragraph: ω is used before introduction

- Page 11 and 12: the reader is left alone with the notation $\rho_{2+/-}$. It would improve the readability of the manuscript if the authors would explain briefly what does this notation physically capture.

- Page 13: "increase the density of infected" of what?

Reviewer #2 (Remarks to the Author):

Simplicial complexes have gained much attention in the data mining community. Among those systems that have been investigated as simplicial complexes, networked systems play an important role. However, a majority of works use tools like filtration as features to extract information about the system, but the implication of complexes on how the system functions is still poorly known. I have ambivalent opinions about this research papers. On the one hand, it proposes a simple, tractable model of dynamics on simplicial complexes. The paper is well-written, clear and I am persuaded that it could open interesting lines of research in the future. On the other hand, the paper suffers from several limitations, and the authors will have to address them for this paper to be considered in NatComm.

1. The authors describe the system as a simplicial complex, but the geometric aspect of the model is non-existent. The authors should better argue why simplicial complexes are indeed necessary,

and provide more convincing arguments than « all subsimplexes of a simplex are included ».

2. There have been many works on opinion dynamics on hypergraphs. The authors should better argue why their model is needed with respect to this literature. Am I correct that their model could be mapped on a model with hypergraphs, if the number of hyper edges of different types is carefully chosen?

3. The model has, in principle, many parameters. Importantly, it does not have any empirical basis to motivate the model or calibrate the parameters.

4. Threshold models, as by Watts for instance, appear to provide qualitatively similar properties, and help to understand the importance of a critical mass for dynamics. Could the authors comment?

5. Two of the authors have recently published a different yet philosophically related paper in PRL. It is my belief that this previous publication diminishes the importance of the current submission, but I will be happy that the authors contradict me.

Responses to Referee 1

> Referee 1: I have carefully read the manuscript Iacopini and others entitled as Simplicial models of social contagion, which was submitted to Nature Communications for publications. The paper propose a new model and its solution to consider higher order structural correlations in the dynamics of social contagion phenomena. The authors, being well known experts in the field of simplicial complexes, provide a concept to consider not only dyadic social influence but group influence as well in models of social spreading by introducing higher-order structures using k-simplexes in the network description. After introduction, the authors successfully demonstrate the numerical simulations of their model and its mean-field approximate solution. They show that for increasing higher-order patterns the spreading process reaches a larger density of nodes in its stationary state via a continuous phase-transition, while for larger density of 2-simplices it goes through a discontinuous phase-transition with a bistable region. The paper is well written, addresses an important extension of well known modelling methods, and explains a new phenomena. On the other hand I have a few concerns and comments on the manuscript regarding its model choice, generalisation, while I also argue that some of the claims of the authors, e.g. the seed size dependence of the spreading process, is not evidently supported by the results, thus requires further explanation. Please find my comments below.

Response: We thank the Referee for his/her very positive appreciation of our work. We are grateful for the comments and suggestions. We have taken them into account as described below in details, improving our work by adding more details, explanations and also new investigations and analysis, both analytical and numerical.

> Referee 1: (a) In the abstract the authors mention that their contributions are a first step to understand the role of high-order interactions in complex systems. Considering higher-order correlations in complex systems, especially in complex networks is not entirely new as even some of the authors have extensive published work in this area. Thus, claiming that this manuscript is the first step is certainly an overshoot.

Response: The Referee is right and we have corrected the last sentence of the abstract, stating now simply that our work contributes to the understanding of the role of higher order interactions.

> Referee 1: (b) In the introduction the authors argue the difference between simple and complex contagion assigning the latter one as the model adequate for the modelling of social contagion. Complex contagion is modelled by threshold driven mechanisms [see definition by Centola & Macy 2007], which capture accumulated social influence what individuals mount towards their threshold of adoption. Complex contagion processes are deterministic in their simplest definition and adoption is not driven by any probabilistic rate of transmission. Contrary to this picture, the authors model social contagion with an SIS type of process, where transition between node states are stochastic and driven by probabilistic rates. The model used in this manuscript actually extends the SIS model with group infection, which is yet driven by a probabilistic rate and recovery and has nothing to do with thresholds and complex contagion. SIS model is a simple contagion process, meant originally to describe biological epidemics, and certainly may have some relevance in social contagion. So I would suggest for the authors to make this point clear in the introduction otherwise they blur their contributions between these two modelling paradigms.

Response: We apologize for the lack of clarity in the introduction. Centola and Macy define a

contagion as complex "if its transmission requires an individual to have contact with two or more sources of activation", and different models have been proposed to take this point into account. As indicated by the referee, a very popular modeling avenue is given by threshold-based models in which transitions are deterministic. The most famous model of this type is probably the one by Watts, and a number of authors have proposed extensions of it, still with deterministic dynamics. On the other hand, there is another modeling avenue for social contagion, given by "epidemic-like social contagion", in which the effect of multiple exposures is taken into account in some works [20,22] while simple contagion is considered in other cases [19,21]. Our work collocates itself in this framework, as noted by the Referee, but using in fact a superposition of simple contagion processes (given by the pairwise interactions) and complex ones, given by the group interactions in which contagion occurs, as defined by Centola and Macy, if the susceptible individual is in "contact with two or more sources" of contagion. We have modified the introduction in order to discuss the various types of modeling frameworks and to make clear that our contribution puts forward an epidemic-like social contagion model.

> **Referee 1: (c) On page 4 the authors introduce the concept of a k -simplex as "We recall that a k -simplex σ is in its simplest definition a collection of $k - 1$ vertices $\sigma = [p_0, \dots, p_{k-1}]$ ". In my understanding a k -simplex is the collection of $k + 1$ vertices. This is also compatible with their examples on page 5 where they call 0-simplex a node, 1-simplex an edge, and 2-simplex a triangle.**

Response: We thank the referee for pointing it out. This was a mistake, which has now been fixed in the updated version of the manuscript.

> **Referee 1: (d) Characterising social groups with higher order simplices might be a too rigid approach due to the restrictive geometric definition of a simplex. This way I understand why the authors only considered 1 and 2 simplices as building blocks of a social network. On the other hand, getting away from the social concept (which I understood might be out of the scope of the paper) it would very interesting to see how the critical behaviour of an SIS process generalise for higher D values.**

Response: We agree with the Referee on the interest of higher D values, and in particular of checking whether the phenomenology we observe for $D = 2$ is still present. While a full solution of the general D case remains beyond reach, and would yield a phase diagram with too many parameters, we have nonetheless considered two specific cases in which analytical results can be obtained (at the mean-field level), and we present them in the new Supplementary Note 3. We first tackle the case $D = 3$, with for simplicity $\beta_2 = 0$ (β_2 is the parameter called also β_Δ in the main text). We show that, for $\lambda = \beta_1/\mu$ less than 1, there exists a discontinuous transition when β_3 increases, similar to the discontinuous transition discussed in the main text. We then consider the case of a general D but with all parameters $\beta_1 = \dots = \beta_{D-1}$ equal to 0. We can here also show analytically that there exists a discontinuous transition as β_D increases.

The fact that these two cases yield discontinuous transitions similar to the one of $D = 2$ gives us confidence that it is a general phenomenology holding for general $D > 2$. We have added a comment in the main text on these additional results.

> **Referee 1: (e) On page 8, in the approximate definition of the average degree of their model network does not consider links which simultaneously appear via both link addition process (by adding random links and triangles). This may set off their MF approximation results, as the networks they consider has relatively high degree, thus the contribution of multiple links might not be negligible. The subtraction of double counted links might be easy and may improve the fit between numerical simulations and the mean-field curves e.g. on Fig.2.**

Response: We thank the referee for this suggestion. We have measured the fraction of overlapping

links for the considered parameters: it is roughly 0.003, i.e. very small. In any case, we adjusted the approximation for the average degree considering the previously created links when counting the ones created by the introduction of 2-simplices. We have then checked the accuracy of the expected $\langle k_1 \rangle$ and $\langle k_\Delta \rangle$ by comparing them with the ones obtained by averaging different realizations of the model. These are reported in a Supplementary Note 1 together with the respective generalized degree distributions.

> **Referee 1: (f) It would be interesting to see how the histeresis and the phase-transitions in Fig.2 depends on the system size.**

Response: We have checked the size effects in the behavior of the hysteresis by performing simulations of the SCM on systems of different sizes, namely $N = 500, 1000, 2000,$ and 4000 , while keeping λ_Δ fixed within the region where we observe the bi-stability ($\lambda_\Delta = 2.5$). As the dedicated Supplementary Note 2 in the Supplementary Information shows, we do not observe a significant variation of the dynamics when simplicial complexes of different sizes are considered, apart from a general stabilization of the incidence curves whose fluctuations tend to be smaller as the size increases.

> **Referee 1: (g) Page 12 and 13: The authors write that “the final state depends on the initial density of infectious for $\lambda_c < \lambda < 1$, that is, a critical mass is needed to reach the endemic state”. First of all “infectious” of what? Second of all, I cannot find any evidence in their results suggesting this conclusion. It would be necessary to explain this more in details by referring precisely which details of which figures support this claim, or remove it from the manuscript and also from the abstract.**

Response: We apologize for the fact that these results were not explained clearly enough. We have developed in more details the analysis of the steady states of the evolution of the density of infectious nodes, and the analysis of their stability in the various regimes.

In particular, in the regime $\lambda_c < \lambda < 1$ (for $\lambda_\Delta > 1$), we obtain three solutions with $\rho_1^* = 0 < \rho_{2-}^* < \rho_{2+}$. As a result, both $\rho_1^* = 0$ and ρ_{2+} are stable, while ρ_{2-}^* is unstable: indeed, the evolution equation shows that $d_t \rho$ is negative for $\rho(t)$ between 0 and ρ_{2-}^* , and positive for $\rho(t)$ between ρ_{2-}^* and ρ_{2+}^* . Thus, depending on the initial value $\rho(t=0)$, $\rho(t)$ will tend either to 0 or to the positive stable state ρ_{2+}^* .

In addition, we show now also numerical evidence that the long time limit of $\rho(t)$ depends on its initial value in Figures 3a and 3b. First, we show in Figure 3a $\rho(t \rightarrow \infty)$ (averaged over runs) as a function of the parameter λ for two values of the initial density of infectious nodes, for $\lambda_\Delta = 2.5$: in a certain range of λ values, $\rho(t \rightarrow \infty) = 0$ for $\rho(t=0) = 0.01$, while $\rho(t \rightarrow \infty) > 0$ for $\rho(t=0) = 0.4$ (see also Supplementary Note 2). Note that, for $\lambda_\Delta = 0.8$, no such dependence on the initial condition is observed. Second, we show in Figure 3b the time evolution of the density of infectious nodes $\rho(t)$ for single runs with varying initial density of infectious nodes, for $\lambda = 0.75$ and $\lambda_\Delta = 2.5$. It is clearly seen that the density of infectious nodes goes to 0 if it is initially below ρ_{2-}^* , and goes instead to a finite value if it starts above ρ_{2-}^* , as predicted by the mean-field analysis.

> **Referee 1: (h) In the conclusion the authors mention in the last sentence that their modelling might be important to distinguish between higher-order dynamic effects in real data. They should discuss this more in details, maybe giving examples how their method can be applied on data. Without this, the last sentence is irrelevant. Actually, one of the main shortcoming is a missing data-driven simulation study on a small real social networks with a simulated SIS process.**

Response: We have modified the last sentence of the article to tone down our claim. In addition, we have added a simulation study on several real-world datasets. In particular, we have consid-

ered four co-presence data sets that provide high-resolution face-to-face interactions as recorded by RFID tags placed on the chest of participants in different social contexts: a workplace, a conference, a hospital and a high school. The results of the simulations of the simplicial contagion model on the clique-complexes constructed on top of the data sets are now presented as first numerical exploration of the SCM. All the details on the data aggregation and processing and augmentation are explained in the new dedicated “Methods” section. The results show the robustness of the observed phenomenology, with appearance of a discontinuous transition and of a hysteresis loop at large enough λ_Δ .

> **Referee 1: Minor Points:**

1. **Page 7: Typo at “in which both j and k are infectious, ,”**
2. **On page 9 and later, calling a phase transition explosive suggest a certain mechanisms (present in explosive percolations), which is not characterising the actual system at all. I would suggest to call it discontinuous or 1st order phase transition.**
3. **Page 11, 1st paragraph: ω is used before introduction**
4. **Page 11 and 12: the reader is left alone with the notation $\rho_{2+/-}$. It would improve the readability of the manuscript if the authors would explain briefly what does this notation physically capture.**
5. **Page 13: increase the density of infected of what?**

Response: We have corrected the typos, defined ω , rewritten the definitions of ρ_{2+} and ρ_{2-} , and replaced ”explosive” by discontinuous.

Responses to Referee 2

> **Referee 2:** Simplicial complexes have gained much attention in the data mining community. Among those systems that have been investigated as simplicial complexes, networked systems play an important role. However, a majority of works use tools like filtration as features to extract information about the system, but the implication of complexes on how the system functions is still poorly known. I have ambivalent opinions about this research papers. On the one hand, it proposes a simple, tractable model of dynamics on simplicial complexes. The paper is well-written, clear and I am persuaded that it could open interesting lines of research in the future. On the other hand, the paper suffers from several limitations, and the authors will have to address them for this paper to be considered in NatComm.

Response:

We are glad that the Referee found the paper well written and considers that it could open interesting lines of research in the future. We are also grateful for his/her comments, which have given us the possibility to clarify some important points and cite relevant literature. We have considered his/her concerns as detailed below, and we hope to have been able to satisfactorily address all of them.

> **Referee 2: (a)** The authors describe the system as a simplicial complex, but the geometric aspect of the model is non-existent. The authors should better argue why simplicial complexes are indeed necessary, and provide more convincing arguments than “all subsimplexes of a simplex are included”.

Response: We thank the Referee for addressing this important point. We have now provided more explanations in the introduction to justify our modeling assumption that all subsimplexes of a simplex are included when describing higher-interactions in social systems. Note that this assumption also amounts to use simplicial complexes instead of the more general framework of hypergraphs: this point is thus linked to the point (b) below on the relations between models on hypergraphs and on simplicial complexes.

The revised version of the manuscript now reads: *“with the extra requirement that if simplex $\sigma \in \mathcal{K}$, then all the sub-simplices $\nu \subset \sigma$ built from subsets of σ are also contained in \mathcal{K} . Such a requirement, which makes simplicial complexes a special kind of hypergraphs (see Supplementary Note 4), seems to be appropriate in the definition of higher-dimensional groups in the context of social systems, and simplicial complexes have been used to represent social aggregation in human communication [Kee et al. (2013)]. Removing this extra requirement would imply, for instance, modelling a group interaction of three individuals without taking into account the dyadic interactions among them. The same argument can be extended to interactions of four or more individuals: it is reasonable to assume that the existence of high-order interactions implies the presence of the lower-order interactions.”*

> **Referee 2: (b)** There have been many works on opinion dynamics on hypergraphs. The authors should better argue why their model is needed with respect to this literature. Am I correct that their model could be mapped on a model with hypergraphs, if the number of hyper edges of different types is carefully chosen?

Response:

Hypergraphs are a generalization of the concept of graphs in which the edges, called hyperedges, can join any number of vertices. Formally, a hypergraph \mathcal{H} is the pair of sets (V, E) , where V is a set of vertices, and the set of hyperedges E is a subset of the power set $P(V)$ of V . Simplicial

complexes are therefore, as observed by the Referee, special kinds of hypergraphs, which contain all subsets of every hyperedge. A simplicial complex \mathcal{K} on the set of vertices V can indeed be seen as a hypergraph \mathcal{H} on V if the latter satisfies the extra requirement that, for each $\sigma \in E$, and for all $\nu \neq \emptyset$ such that $\nu \subseteq \sigma$, we also have $\nu \in E$.

Such an extra requirement seems appropriate in the context of models of social interactions considered in our work, and we have added arguments to justify this choice in the introduction (see also answer to point (a) above). Removing this extra requirement would imply, for instance, modelling a group interaction of three social individuals without taking into account also the dyadic interactions among them. The same argument can be extended to interactions of four or more individuals: the existence of high-order interactions implies the presence of the lower-order interactions. Of course, the relative importance and contributions of the lower-order interactions can be controlled in our model of social contagion by tuning the parameters $\beta_1, \beta_2, \beta_3$, and so on.

The Referee is also correct in writing that our model can be mapped on a model with hypergraphs if the hyperedges of different types are carefully chosen. However, in addition to the point above on the representation of social interactions, using simplicial complexes has the advantage to keep the model relatively simple, with a controlled number of parameters, and amenable to analytical solution.

We have now added a comment in the main text (in the conclusion) about this point: *“Furthermore, given that the SCM can be mapped on a model with hypergraphs if the hyperedges of different types are carefully chosen, it would be interesting to study the behavior of complex contagion processes on more general classes of hypergraphs.”*, and Supplementary Note 4 on the relations between hypergraphs and simplicial complexes. We have also added references to two papers [Lanchier, Neuffer 2013] and [Bodó et al., 2016] on dynamical processes (respectively opinion dynamics and SIS); on hypergraphs. The motivations, focus and results of these two works are however very different from ours.

> **Referee 2: (c) The model has, in principle, many parameters. Importantly, it does not have any empirical basis to motivate the model or calibrate the parameters.**

Response: We have added motivation for the model in the now extended introduction and provided additional context to justify the interest in our model. We agree with the referee that the problem of inferring the values of the parameters (the various β .) from real world data is an important direction for further research, particularly in the direction of teasing apart the contributions of the various orders. This is however a hard inference problem which requires high-resolution data of both the structural substrate and of the dynamics evolving on it. This is certainly in our scopes for future work, but we believe it goes beyond the scope of this paper.

> **Referee 2: (d) Threshold models, as by Watts for instance, appear to provide qualitatively similar properties, and help to understand the importance of a critical mass for dynamics. Could the authors comment?**

Response: Threshold models are indeed one of the well-known avenues for modeling social contagion phenomena, and show for instance the importance of the initial conditions. However, such models describe cascades and reach usually a frozen state in which no dynamics occur anymore. Our work lies rather in the framework of “epidemic-like” social contagion models, in which non-equilibrium steady states can be reached. As also described in one of the answers to Referee 1’s comments, we have substantially rewritten the introduction to make these points clear.

> **Referee 2: (e) Two of the authors have recently published a different yet philosophically related paper in PRL. It is my belief that this previous publication diminishes the importance of the current submission, but I will be happy that the authors contradict me.**

Response: Two of the authors have indeed recently published a paper titled "Simplicial activity driven model". This paper however is totally unrelated, except for the fact that it also considers simplicial complexes. Indeed, the focus in the PRL paper is the definition of a new model for a temporally evolving structure: the paper studies the resulting structural properties and some well-known processes on top of it, without introducing any new model of processes.

On the contrary, the focus of the present manuscript is on a new model for a contagion process, i.e., a new model for a process that takes place on any fixed, non-evolving structure.

In other words, the PRL paper deals with a model of a time-varying structure, while our manuscript deals with processes taking place on a fixed structure. In some sense, the difference between the PRL paper and our new manuscript is of the same type as between a paper describing a new (temporal) network model and a paper describing a new model of processes on top of generic networks.

REVIEWERS' COMMENTS:

Reviewer #1 (Remarks to the Author):

After carefully reading the revised manuscript I can confirm that the authors addressed all my comments thoroughly and the manuscript improved considerably. I very much liked the results on the generalisation of their results for higher dimensions (Supp. Note 3) and the new data-driven simulations what they carried out on small RFID datasets. These results underline the generality and applicability of their results, and opens the perspective of the paper to be interesting for a broader scientific audience.

Reviewer #2 (Remarks to the Author):

Dear Editor,

The authors have addressed all my comments. In my view, the manuscript is now much more comprehensive and provides better arguments about the novelty and potential impact of the work. The weak side remains the fact that model parameters are not data-driven, which implies that any behaviour could actually emerge from the proposed process. Nonetheless, this generality also opens interesting research perspectives and provides sound basis for future research. For these reasons, my recommendation is to accept this paper for publication Nature Com.

Responses to Referees

> Referee 1: After carefully reading the revised manuscript I can confirm that the authors addressed all my comments thoroughly and the manuscript improved considerably. I very much liked the results on the generalisation of their results for higher dimensions (Supp. Note 3) and the new data-driven simulations what they carried out on small RFID datasets. These results underline the generality and applicability of their results, and opens the perspective of the paper to be interesting for a broader scientific audience.

> Referee 2: The authors have addressed all my comments. In my view, the manuscript is now much more comprehensive and provides better arguments about the novelty and potential impact of the work. The weak side remains the fact that model parameters are not data-driven, which implies that any behaviour could actually emerge from the proposed process. Nonetheless, this generality also opens interesting research perspectives and provides sound basis for future research. For these reasons, my recommendation is to accept this paper for publication Nature Com.

Response: We thank both referees for their careful review of our work and the very positive evaluation received. Their precious comments and constructive criticisms led to a much improved manuscript.